# DNA Methylation in Nonalcoholic Fatty Liver Disease

**DOI:** 10.3390/ijms21218138

**Published:** 2020-10-30

**Authors:** Jeongeun Hyun, Youngmi Jung

**Affiliations:** 1Institute of Tissue Regeneration Engineering (ITREN), Dankook University, Cheonan 31116, Korea; j.hyun@dankook.ac.kr; 2Department of Nanobiomedical Science and BK21 PLUS NBM Global Research Center for Regenerative Medicine, Dankook University, Cheonan 31116, Korea; 3Department of Biomaterials Science, College of Dentistry, Dankook University, Cheonan 31116, Korea; 4Cell and Matter Institute, Dankook University, Cheonan 31116, Korea; 5Department of Integrated Biological Science, Pusan National University, Pusan 46241, Korea; 6Department of Biological Sciences, Pusan National University, Pusan 46241, Korea

**Keywords:** nonalcoholic fatty liver disease (NAFLD), nonalcoholic steatohepatitis (NASH), DNA methylation, epigenetics, cytosine-phospho-guanine dinucleotide (CpG), biomarker, 5-methylcytosine (5-mC), 5-hydroxymethylcytosine (5-hmC), DNA methyltransferase (DNMT), ten eleven translocation (TET)

## Abstract

Nonalcoholic fatty liver disease (NAFLD) is a widespread hepatic disorder in the United States and other Westernized countries. Nonalcoholic steatohepatitis (NASH), an advanced stage of NAFLD, can progress to end-stage liver disease, including cirrhosis and liver cancer. Poor understanding of mechanisms underlying NAFLD progression from simple steatosis to NASH has limited the development of effective therapies and biomarkers. An accumulating body of studies has suggested the importance of DNA methylation, which plays pivotal roles in NAFLD pathogenesis. DNA methylation signatures that can affect gene expression are influenced by environmental and lifestyle experiences such as diet, obesity, and physical activity and are reversible. Hence, DNA methylation signatures and modifiers in NAFLD may provide the basis for developing biomarkers indicating the onset and progression of NAFLD and therapeutics for NAFLD. Herein, we review an update on the recent findings in DNA methylation signatures and their roles in the pathogenesis of NAFLD and broaden people’s perspectives on potential DNA methylation-related treatments and biomarkers for NAFLD.

## 1. Introduction

Nonalcoholic fatty liver disease (NAFLD) is a common hepatic disorder occurring in individuals who drink little or no alcohol. NAFLD is becoming a global health problem and currently affects one-fourth of the general population worldwide [1]. The obesity epidemic has increased the prevalence of NAFLD, which is also closely related to hyperlipidemia and metabolic syndrome (MetS) [1,2]. The majority of patients with NAFLD have simple steatosis, which refers to an excessive amount of lipids (more than 5% of the liver weight) accumulated in the liver [3]. However, a subset of patients with NAFLD progress to a severe form of liver disease called nonalcoholic steatohepatitis (NASH) in which inflammation, hepatocyte ballooning and fibrosis are evident. Importantly, NASH can advance to end-stage liver disease, such as cirrhosis and liver cancer [4,5,6,7]. NASH has overtaken the impact of hepatitis B and C and has become a principal cause of liver disease-related death [8]. In the United States, NASH is expected to be the leading indication for liver transplantation in the coming decade [8,9].

Given the increasing prevalence and progression of NAFLD, it is important to discriminate between patients at high risk and those at low risk for the development of end-stage liver disease. In other words, it is key to differentiate NASH from simple steatosis in the NAFLD population. It is also important to identify the severity of hepatic fibrosis, since advanced fibrosis has been shown to be the major driver for long-term outcome and mortality [10,11,12]. However, the risk of disease progression is highly variable among patients, and the underlying mechanisms are poorly understood, which has limited the development of effective strategies for treating and staging NAFLD.

There is strong evidence that personal environment and lifestyle impact gene expression through numerous epigenetic mechanisms, influencing the phenotype and behavior of cells and tissues [13,14]. The epigenetic processes including DNA methylation, histone modifications and non-coding RNAs have been reported to be related to the molecular basis of liver diseases including NAFLD [15,16]. This may help explain inter-individual variabilities in disease susceptibility and outcome [17,18,19]. Of these epigenetic processes, DNA methylation is a fundamental epigenetic modification of DNA that occurs at the cytosine base within a cytosine-phospho-guanine (CpG) dinucleotide and is closely associated with transcription factor binding and chromatin accessibility [20,21,22,23]. In the past, CpG methylation was thought to be stable in the genome [24]. With the development of next-generation sequencing that allows quantification of DNA methylation at the single-base resolution, however, it has been revealed that DNA methylation is highly dynamic and changes in response to cellular and tissue microenvironments [25,26]. It has been reported that the modification of DNA methylation occurs during the growth and spread of a tumor and chemotherapy-received tumors [27], and it is mechanistically implicated in the activation of hepatic stellate cells (HSCs) [25,28,29]. Since genome-wide association studies have been applied in studies on human liver diseases [30,31], there is the potential to detect epigenetic modifiers in NAFLD, which might provide therapeutic targets and molecular tools for diagnosing NAFLD, assessing the severity, and predicting disease progression [32].

As researchers are increasingly aware of the importance of DNA methylation in liver pathogenesis, a number of publications on DNA methylation in liver disease have dramatically accumulated in the last two decades ago. Although the absolute number of publications on DNA methylation, particularly in NAFLD or NASH, is small, studies are expanding (Figure 1). In this review, we discuss the current status of studies focusing on the identification of NAFLD-associated global and locus-specific DNA methylation and aim to urge more and further well-designed studies in this field.

## 2. Histological Features of NASH

NASH can be distinguished from simple steatosis by the histological characteristics of “steatohepatitis”, including steatosis, lobular inflammation, and hepatocyte ballooning with or without peri-sinusoidal fibrosis.

### 2.1. Steatosis

Hepatic steatosis means that fat accumulates that accounts for more than 5% of the liver weight [3]. Fat begins to accumulate in hepatocytes around the terminal hepatic venule, and a single or a few macrovesicular lipid droplets are predominantly observed in NAFLD. Microvesicular steatosis has been shown to occur in up to 10% of NASH cases and to be related to ballooned hepatocytes and advanced fibrosis [33,34].

### 2.2. Hepatocyte Ballooning

Hepatocyte ballooning indicates the degeneration of hepatocytes (i.e., microtubular disruption and severe hepatocyte injury) that appear morphologically as enlarged and swollen hepatocytes [35]. Ballooned hepatocytes are typically seen around terminal hepatic venules in adult NASH, and they are mingled with perisinusoidal collagen fibers in the fibrotic liver. The loss of cytokeratin 8 and 18 (CK8/18) is observed in ballooned hepatocytes, whereas CK8/18 is present in both the membrane and cytoplasm of normal hepatocytes [36]. In ballooned hepatocytes, CK8/18 highlights Mallory-Denk bodies (MDB) that are aggregates of ubiquitinated keratins within the proteasome [37]; therefore, double immunostaining for CK8/18 and ubiquitin has been used for the detection of hepatocyte ballooning in NASH [38]. p62/Sequestosome-1 is also a marker of MDB [39]. It has been reported that hepatocyte ballooning is associated with insulin resistance (IR) [38], increased serum cholesterol [38,40], and necroinflammation [41].

### 2.3. Inflammation

In adult NAFLD, lobular inflammation is more distinct than portal inflammation. Lobular inflammation consists of Kupffer cell clusters without (microgranulomas) or with lipid droplets (lipogranulomas), mononuclear cells, and occasional polymorphonuclear leukocytes [42]. In NASH, Kupffer cells are enlarged and aggregated around terminal hepatic venules whereas they are distributed evenly in normal livers and in simple steatosis [43]. Portal inflammation that typically consists of mononuclear cells [44] is known to be associated with increased steatosis, ballooning and fibrosis in both adult and pediatric NASH [45].

### 2.4. Fibrosis

Pericellular fibrosis located around the terminal hepatic venule is typical in adult NASH [3]. As NAFLD progresses, the fibrosis lining the perisinusoidal space becomes thicker, and portal and periportal fibrosis can be formed often alongside the ductular reaction, which refers to an increased number of ductular cells [46]. Eventually, bridging fibrosis is formed, hepatic architecture is remodeled and, in the end, cirrhosis may develop [47].

## 3. Pathway of DNA Methylation

### 3.1. DNA Methylation Pathways: DNA Methyltransferases (DNMTs)

DNA methylation is the major epigenetic modification in eukaryotic cells and involves the incorporation of a methyl group on the fifth carbon pyrimidine ring in cytosine (C) nucleotides to generate 5-methylcytosine (5-mC) [20,48]. It is estimated that 1% of the DNA bases are 5-mC, which accounts for 70–80% of CpG dinucleotides [48]. Methylated cytosines are mainly located in long stretches of DNA containing highly dense CpG clusters called CpG islands. CpG islands are mostly found in regulatory or promoter regions of the genes closely related to the transcription start site [49,50,51] in the unmethylated state, whereas many nonpromoter CpGs are methylated throughout the genome [52]. Cytosine methylation is a stable and inherited epigenetic mark and usually represses transcription, either by preventing the binding of transcription factors or by recruiting methyl-CpG-binding domain proteins, such as methyl CpG binding protein 2 (MeCP2), methyl-CpG-binding domain 1-4 (MBD1-4) and Kaiso, to methyl groups, thereby impacting on cell/tissue-specific gene expression [20,24,48,49,52,53] (Figure 2). Moreover, cytosine methylation influences chromatin structure and participates in numerous processes, such as inactivation of the X chromosome, genomic imprinting, embryogenesis, and cellular differentiation [54].

DNA methylation is conducted by DNA methyltransferases (DNMTs) that catalyze the transfer of a methyl group from S-adenosyl methionine (SAM) to cytosine [55] (Figure 3). There are four members of the DNMT family in human, DNMT1, DNMT3A, DNMT3B and DNMT3L, and DNMT1, DNMT3A and DNMT3B among them have been identified to have DNA methyltransferase activities [52]. DNMT1 has a unique ability to copy CpG methylation patterns and add them to the newly synthesized DNA strand, maintaining DNA methylation status during DNA replication [49,56]. Both DNMT3A and DNMT3B are de novo DNA methyltransferases [57,58]. Although there is high homology between DNMT3A and DNMT3B, they have different expressional patterns and targets. DNMT3A is ubiquitously expressed in the late stages of embryogenesis and the differentiated cells, while DNMT3B is dominant in the early stages of embryogenesis [49,59,60]. In addition, DNMT3A and DNMT3B cooperate with DNMT1 in the maintenance of DNA methylation [49]. DNMT3L has high homology with DNMT3A and DNMT3B, but has no methyltransferase activity.

### 3.2. DNA Demethylation Pathways: Ten Eleven Translocation Enzymes (TETs)

Recently, an old modification of mammalian DNA [61], 5-hydroxymethylcytosine (5-hmC), was rediscovered as an epigenetic modification that regulates gene transcription by influencing putative DNA demethylation and chromatin structure remodeling [62,63,64]. In contrast to 5-mC that is mainly found in heterochromatin, 5-hmC is enriched over the bodies of expressed genes and enhancers [65,66,67,68]. Interestingly, the 5-hmC expression is tissue-specific [65] and highly dependent on the cellular state, resulting in epigenetic changes in response to environmental stimuli and metabolic perturbations [66]. In addition, 5-hmC influences both activation and inhibition of transcription depending on the cellular and molecular environment in which it is present [69]. In the adult liver, 5-hmC is abundant in genes involved in active catabolic and metabolic processes [68].

5-hmC is generated by oxidation of 5-mC by ten-eleven translocation (TET) enzymes, a group of Fe^2+^-dependent dioxygenases [63,70,71]. These enzymes are recently identified as 5-mC hydroxylases and use α-ketoglutarate as a co-substrate in 5-mC oxidation (Figure 3). Because this reaction is important in the DNA demethylation process, these enzymes are considered as the potential regulators for epigenetics [63]. In mammals, there are three types of TET proteins, TET1, TET2, and TET3, and their expressional types are different depending on cells/tissues. TET1, the most studied enzyme among them, initiates oxidation in DNA demethylation process in mammals [70,72].

## 4. DNA Methylation in NAFLD

### 4.1. Animal Models

The methyl-group donors from foods, such as folate, betaine and choline, are required for SAM synthesis [73,74]. Since DNA methylation relies on the availability of SAM, diet is one of the main factors that influence DNA methylation. Diets depleted of methyl donors decreased the level of hepatic SAM and caused CpG island demethylation of 164 genes in mouse livers [75]. These genes were shown to be involved in DNA damage/repair, lipid and glucose metabolism, and fibrogenesis. Folate is a catalytic substrate for transferring one-carbon units. It was reported that folate affected the expression of genes regulating fatty acid synthesis, and folate deficiency-induced triglyceride (TG) accumulation in the liver [76,77]. Betaine has also been found to relieve high-fat diet (HFD)-induced fatty liver [78]. Wang et al. [79] showed that supplementation with betaine reduced methylation at the promoter region of the microsomal triglyceride transfer protein (*Mttp*) gene in mice and induced global methylation over the genome. This promoted hepatic TG export and attenuated liver steatosis in mice with HFD-induced NAFLD. Methyl donor supplementation containing folic acid (a synthetic version of folate), betaine, choline and vitamin B12 also reverted hepatic fat accumulation in high-fat and high-sucrose (HFS) diet-fed rats by affecting the DNA methylation patterns in the promoter regions of sterol regulatory element-binding protein 2 (*Srebf2*), 1-acylglycerol-3-phosphate O-acyltransferase 3 (*Agpat3*) and estrogen receptor 1 (*Esr1*) genes, which are involved in obesity development and lipid metabolism, and modifying their mRNA expression [80]. Similarly, methyl donor supplementation induced fatty acid synthase (*Fasn*) DNA hypermethylation, which may also mediate the improvement in HFS-induced NAFLD [81].

Although the above studies show that NAFLD is reversible by changing the diet, Kim et al. [82] recently found that elevated serum TG levels persisted in HFD- or high-fructose diet (HFrD)-fed mice for 9 weeks after changing to a normal chow diet for an additional 9 weeks. The metagenomic analysis demonstrated that some microbiome compositions (e.g., *Odoribacter* genus) changed by HFD or HFrD remained altered even after reversing the diet to normal chow. In addition, DNA hypomethylation of key liver genes involved in lipid metabolism (e.g., *Apoa4*) was revealed by genome-wide DNA methylation analysis to remain in both diet-fed groups even after introduction to a normal chow diet. This study indicates that changes in microbiome composition may contribute to perpetual epigenetic modifications in the liver. This “priming effect” through changes in DNA methylation is further supported by increasing evidence showing that the intrauterine environment affects later life in animal models [83,84,85,86,87]. Given that DNA methylation is inheritable [88], the early postnatal offspring from HFD-exposed maternal mice can have hepatic dysfunction [89]. In maternal HFD offspring, cyclin-dependent kinase inhibitor 1A (*Cdkn1a*), a cell cycle inhibitor, was hypomethylated, and CDKN1A expression was upregulated, which in turn correlated with hepatocyte growth suppression. Recently, a similar study has shown that a high-fat and high-cholesterol Western diet (WD)-induced maternal hypercholesterolemia predisposed offspring to NAFLD and metabolic diseases [90]. Female apolipoprotein (Apo) E-deficient mice were fed WD before and throughout pregnancy and lactation, and their pups were weaned onto a normal chow diet after birth. Male offspring exposed to high-cholesterol in utero had glucose intolerance, peripheral insulin resistance and hepatic steatosis at 4 months of age. These NAFLD-like phenotypes were, at least in part, related to increased methylated CpG sites on the promoter region of the *ApoB* gene in the livers of male offspring of WD-fed mice compared with male mice born to chow-fed mice. Hypermethylation of the *ApoB* gene that is required for very-low-density lipoprotein (VLDL) cholesterol assembly resulted in lower expression of hepatic ApoB, contributing to the changes in the lipid profile. Human epidemiological studies have also demonstrated that detrimental environment during prenatal and early postnatal periods have long-term effects on the development of NAFLD-related clinical disorders [91], including insulin resistance [92], abdominal obesity [93], diabetes [94], hyperlipidemia [95], cardiovascular disease [96], and metabolic syndrome [97] in adulthood.

There are more studies revealing the association between DNA methylation and NAFLD pathogenesis in rodent models. Dipeptidyl peptidase 4 (DPP4), an adipokine released by hepatocytes [98], is known to be upregulated in the livers of patients with obesity and NAFLD [99,100,101]. In mice fed an HFD for 6 weeks, hepatic DPP4 expression was increased with high weight gain independent of liver fat content, and the methylation of four intronic CpG sites flanking exon 2 was decreased [98]. In older mice fed an HFD for 13 weeks, the hepatic TG content was elevated only in individuals with upregulated *Dpp4* expression. Additionally, in human liver biopsy specimens from obese patients, DPP4 expression was positively correlated and DNA methylation was negatively correlated with stages of hepatic steatosis and NASH. These data indicate that hepatic *Dpp4* expression is promoted by demethylation of the *Dpp4* gene early in life, which might contribute to an early decline in liver function and later lead to hepatic steatosis. The expression and activity of nuclear factor-erythroid 2-related factor-2 (NRF2), a transcription factor with NAFLD-protective function via the negative regulation of genes promoting lipid accumulation, have been shown to be reduced in the liver with histological criteria of NASH [102]. Hosseini et al. [103] investigated the impact of resveratrol (trans-3,5,4′-trihydroxystilbene), a natural polyphenol compound that has the ability to modulate epigenetic patterns, on the promoter methylation of the *Nrf2* gene and lipid accumulation in mice treated with HFD [104]. Resveratrol attenuated DNA methylation at the *Nrf2* promoter region in the livers of mice fed an HFD, and this effect was correlated with a reduction in TG levels and the expression of lipogenesis-related genes, such as fas cell surface death receptor (*Fas*) and sterol regulatory element-binding protein 1 (*Srebp-1c*). Recently, Lyall et al. [105] first performed a genome-wide analysis of hepatic 5-hmC patterns in a mouse model of HFD-induced obesity. This study found that global 5-hmC levels had no difference, but 5-hmC patterns were changed at functionally important genes in NAFLD pathogenesis. The HFD-induced genes (e.g., cytochrome P450 17A1 (*Cyp17a1*)) gained 5-hmC, whereas the HFD-suppressed genes (e.g., insulin-like growth factor binding protein 2 (*Igfbp2*)) lost 5-hmC. Moreover, these alterations were reversible with weight loss, which supports the data from human studies showing that NASH-associated changes in DNA methylation could be resolved with weight loss following bariatric surgery [31]. Moreover, Komatsu et al. [106] reported DNA hypomethylation in the secreted phosphoprotein 1 *(Spp1*) gene, which is associated with liver fibrosis in mouse livers at early-stage liver fibrosis, suggesting that DNA hypomethylation of the gene precedes the onset of liver fibrosis. Global DNA demethylation during the activation of HSCs was also reported by comparing DNA methylation between quiescent and early culture-activated HSCs by an in vitro experiment [107]. A genome-wide analysis confirmed an integral correlation between promoter methylation and gene expression in culture-activated human HSCs [108].

Additionally, it was recently reported by Borowa-Mazgaj et al. [109] that gradual DNA hypermethylation at the promoter region in the glycine N-methyltransferase (*Gnmt*) gene decreased its gene expression in the livers of three different mouse models of NAFLD and NAFLD-derived hepatocellular carcinoma (HCC): a diet-induced animal model of NAFLD (DIAMOND), a Stelic animal model of nonalcoholic steatohepatitis-derived HCC (STAM), and a choline- and folate-deficient (CFD) diet model. They demonstrated that *Gnmt* downregulation was an early event in NAFLD pathogenesis in mice, and one of the consequences of GNMT inhibition in a human HCC cell line was an increase in genome-wide DNA methylation promoted by an elevated level of SAM.

### 4.2. Human Studies

#### 4.2.1. Liver Tissue (Hepatic DNA)

Aberrant DNA methylation patterns are associated with inappropriate gene expression and the pathogenesis of human diseases [110,111,112]. A number of studies have demonstrated altered DNA methylation profiles in liver biopsy samples collected from individuals who were diagnosed with NAFLD [30,31,113,114,115,116,117,118,119,120,121,122,123,124,125,126] (Table 1). It was reported that the methylation levels of peroxisome proliferator-activated receptor-gamma (PPARγ) coactivator 1a (*PPARGC1A*), a key regulator of mitochondrial biogenesis [127], and mitochondrial transcription factor A (*TFAM*) in the livers of patients with NAFLD were relevant to the IR phenotype [115]. The mitochondria-encoded NADH dehydrogenase 6 (*MT-ND6*) gene is another target of mitochondrial DNA (mtDNA) methylation during the development of NAFLD [116,117,128]. *MT-ND6* is hypermethylated, which is related to the severity of NAFLD, and its mRNA level is considerably low in patients with NASH compared to the level in patients with simple steatosis [116]. Zeybel et al. [129] performed bisulfite pyrosequencing, a sequencing-by-synthesis method used to quantitatively determine the methylation of individual CpG cytosines from PCR amplicons, to determine differences in DNA methylation of genes known to affect fibrogenesis between liver biopsy tissues from patients with mild (histological fibrosis stages F0–F2) and severe (F3–F4) NAFLD [30]. DNA methylation at particular CpG dinucleotides within the human *PPARα* and *PPARγ* gene promoters was upregulated (i.e., hypermethylation), whereas it was downregulated within the platelet-derived growth factor subunit A (*PDGFA*) gene promoter and transforming growth factor 1 (*TGF1*) exon 1 (i.e., hypomethylation) in severe NAFLD compared with mild NAFLD. In addition, global DNA methylation in liver biopsy specimens from patients with NAFLD was recently evaluated [124]. The level of global DNA methylation was significantly lower in the 47 patients with NAFLD than in the 18 control participants who were non-NAFLD overweight. Remarkably, the level of global DNA methylation in the liver tended to decrease with an increase in hepatic inflammation and fibrosis grade and disease progression. In addition, the patients with NAFLD had a significantly higher serum concentration of the one-carbon metabolite homocysteine, which meant reduced production of SAM compared to that of the control group. This was also correlated with hepatic steatosis grade and disease progression. These results indicate that global hepatic DNA methylation and serum one-carbon metabolite levels are associated with the histological severity of NAFLD.

Since the single-CpG-resolution genome-wide DNA methylation profiling platform Infinium HumanMethylation450 BeadChip (Illumina, Inc., San Diego, CA, USA), which is ideal for screening large sample populations, was developed [130], this method has been introduced for the analysis of various human tissue specimens [113,125]. Ahrens et al. [31] ran the Infinium HumanMethylation450 BeadChip assay on 18 normal controls, 18 obese without NAFLD, 12 obese with hepatic steatosis and 15 obese patients with NASH. Of 476 CpGs differentially methylated between the four phenotypic groups, Ahrens et al. mined nine NAFLD-related genes that code for crucial enzymes in intermediate metabolism and insulin-like signaling, including pyruvate carboxylase (*PC*), ATP citrate lyase (*ACLY*), phospholipase C gamma 1 (*PLCG1*), insulin-like growth factor 1 (*IGF1*), *IGFBP2*, protein kinase C epsilon (*PRKCE*), polypeptide N-acetylgalactosaminyltransferase-like 4 (*GALNTL4*), glutamate ionotropic receptor delta type subunit 1 (*GRID1*), and inositol hexakisphosphate kinase 3 (*IP6K3*), and these methylations could be partially reversed after bariatric surgery. Murphy et al. [114] analyzed DNA methylation profiles in frozen liver biopsies from 33 patients with NAFLD with mild fibrosis (F0–F1) and 23 patients with NAFLD with advanced fibrosis (F3–F4). Of 69,247 differentially methylated CpG sites, 76% were hypomethylated and 24% were hypermethylated in patients with NAFLD with advanced fibrosis versus mild fibrosis. Using transcriptome profiling data in the same cohort, it was revealed that DNA methylation correlated with gene transcription levels for 7% of the differentially methylated CpG sites. Murphy et al. [121] found that tissue repair genes, such as fibroblast growth factor receptor 2 (*FGFR2*) and caspase 1 (*CASP1*), were hypomethylated and overexpressed, whereas a gene in one-carbon metabolism, methionine adenosyl methyltransferase 1A (MAT1A), which generates SAM, was hypermethylated and underexpressed in liver biopsies from patients with advanced NAFLD. Methylation at *FGFR2*, *CASP1* and *MAT1A* was validated by bisulfite pyrosequencing, and the findings were reproduced in the replication cohort composed of 19 patients with mild and 15 with advanced NAFLD. Another systemic analysis of DNA methylation and gene expression data from three discrete cohorts including 103 patients with NAFLD and 75 non-NAFLD patients found that 64 genes involved in bile acid homeostasis and drug metabolism were significantly differentially methylated in patients with NAFLD compared to non-NAFLD patients. The methylation levels of 26 out of 64 genes were significantly correlated with their mRNA expression levels, which was NAFLD-dependent. Using the same cohorts, Mwinyi et al. [122] demonstrated that NAFLD and NAFLD-associated fibrosis were associated with significant methylation shifts in 41 genes important for lipid homeostasis and 14 genes involved in energy and vitamin D homeostasis. Pathway analysis showed that Forkhead box a1 (FOXA1) signaling-regulated genes, which are also involved in ultralow-density lipoprotein composition, were affected by changes in methylation and transcription in NAFLD. Moreover, Wegermann et al. [123], in their pilot study involving 86 patients with NAFLD (gene expression and clinical outcome data were available in only 55 patients), found that differential branched-chain amino-acid transaminase 1 (*BCAT1*) gene expression was correlated with changes in DNA methylation at three CpG loci of the *BCAT1* gene in patients with NAFLD who experienced adverse clinical events such as cardiovascular outcomes and/or hepatic decompensation. *BCAT1* is an enzyme that catalyzes the conversion of α-KG to glutamate and has been reported to be related to the presence and severity of NAFLD [131].

DNA methylation is closely associated with genetic polymorphisms in the pathogenesis of NAFLD. For example, DNA methylation in the fatty acid desaturase 1/2/3 (*FADS1/2/3*) gene cluster has been linked with genetic variants and desaturase activities in the human liver [119], and the expression of hepatic *FADS2* mRNA and the activity of serum delta-6 desaturase (D6D) encoded by the *FADS2* gene are known to increase in human NAFLD [132]. Recently, Walle et al. [120] used the Infinium HumanMethylation450 BeadChip to analyze CpG methylation in liver biopsy samples from 95 obese participants and found that the methylation levels in two CpG sites (cg07999042 and cg06781209) annotated to the *FADS2* gene were negatively correlated with hepatic *FADS2* mRNA expression and estimated D6D activity based on both liver and serum fatty acids. Notably, the methylation level of cg07999042 was associated with the *FADS2* variant rs174616.

There is evidence showing that DNA methylation may predict the risk of liver cancer development in patients with NAFLD. Nishida et al. [118] applied a high-throughput quantitative DNA methylation assay that employs fluorescence-based real-time PCR (TaqMan) technology, called MethyLight [133], in liver biopsy samples from 65 patients with NAFLD. Nishida et al. reported abnormal DNA methylation in the promoter regions of 6 tumor suppressor genes (TSGs), including hypermethylated in cancer 1 (*HIC1*), glutathione S-transferase P1 (*GSTP1*), suppressor of cytokine signaling 1 (*SOCS1*), Ras association domain family member 1 (*RASSF1*), cyclin-dependent kinase inhibitor 2a (*CDKN2A*), and adenomatous polyposis coli (*APC*), and found that the level of methylated TSGs was significantly associated with deposition of 8-hydroxydeoxyguanosine (8-OHdG), a marker of oxidative DNA damage, which, in turn, could be a trigger of DNA methylation responsible for hepatocarcinogenesis. In addition, Kuramoto and colleagues [125] analyzed DNA methylation in 264 liver tissue samples composed of 55 normal (NLT), 113 noncancerous NASH (NASH-N), 22 NASH-related HCC (NASH-T), 37 noncancerous chronic hepatitis or cirrhosis associated with hepatitis B or C virus infection (viral-N), and 37 virus-related HCC (viral-T) samples using the Infinium HumanMethylation450 BeadChip. They found 194 distinct CpG methylation sites of NASH-N that were different from those of NLT and viral-N. These altered DNA methylation sites in the NASH-N group from patients without HCC were enhanced in the NASH-N group from patients with HCC and further strengthened in NASH-T samples, indicating that NASH is a precancerous stage for HCC. In particular, tumor-related genes, such as Wolf–Hirschhorn syndrome candidate 1 (*WHSC1*; also known as multiple myeloma SET domain (*MMSET*) or nuclear receptor binding SET domain protein 2 (*NSD2*)), were revealed to be affected by DNA methylation specifically in NASH-N and NASH-T liver tissues, which were not observed in viral-N and viral-T liver samples. Several years later, Tian et al. [126] from the same group reported distinct DNA methylation profiles of 22 NASH-T liver samples different from those of 36 NLT. They found that both DNA methylation and mRNA expression of tumor-related genes, such as tripartite motif-containing 4 (*TRIM4*), protein regulator of cytokinesis 1 (*PRC1*) and tubulin alpha 1b (*TUBA1B*), were altered in NASH-T samples, but not in viral-T, compared to NLT and confirmed that these DNA methylation changes were observed even in NASH-N samples at the precancerous NASH stage. In patients with NASH-T, the alterations in DNA methylation and gene expression of these tumor-related genes were associated with the necroinflammatory grade, which is a characteristic of NASH, and correlated with poorer tumor differentiation, indicating that NASH-specific DNA methylation signatures play a role in NASH-related hepatocarcinogenesis by altering the expression of tumor-related genes.

Taken together, extensive studies have demonstrated that functionally relevant differences in methylation can stratify patients with NAFLD. Genes in which DNA methylation is altered related to the prevalence and progression of NAFLD regulate processes such as steatohepatitis, fibrosis, and carcinogenesis, indicating the role of DNA methylation in the progression of NAFLD and thus its clinical implications for diagnosis, prognosis and therapeutic intervention.

#### 4.2.2. Peripheral Blood or Plasma (Leukocyte DNA or Circulating Cell-Free DNA)

It is believed that dying hepatocytes release degraded genomic DNA systemically, and this cell-free DNA (cfDNA) is easily obtained from plasma [134]. Given that hepatocyte death is the main pathological feature of NAFLD [135], quantification of DNA methylation from peripheral blood or plasma collected from patients could reflect the degree of NAFLD and be suggested as an alternative approach with high accuracy to stratify NAFLD [136] (Table 2). Hardy et al. [136] demonstrated that two specific CpG dinucleotides at the promoter region of *PPARγ*, which is a negative regulator of HSC activation and liver fibrogenesis [28], were hypermethylated in plasma cfDNA isolated from patients with NAFLD with severe fibrosis compared to patients with NAFLD with mild fibrosis. Other studies have also explored the possibility of using peripheral blood leukocyte-derived DNA for the evaluation of potential DNA methylation biomarkers in NAFLD. Nano et al. [137] profiled the DNA methylome in circulating leukocytes from a substantial number of participants (a total of 1450 individuals, including 731 in the discovery cohort and 719 in the replication cohort) to identify DNA methylation sites and levels associated with serum enzyme levels and hepatic steatosis. Among the four genes in which particular CpG methylation was repetitively found to be associated with the serum level of liver enzymes, including solute carrier family 7 member 11 (*SLC7A11*), *SLC43A1*, *SLC1A5* and phosphoglycerate dehydrogenase (*PHGDH*), DNA methylation at the SLC7A11 intronic region (cg06690548) was significantly associated with a reduced risk of hepatic steatosis. Two parallel studies from Dr. Fan’s group investigated DNA methylation in peripheral blood leukocytes from 35 patients with histologically diagnosed NAFLD and 30 healthy participants without liver disease [138,139]. By comparing DNA methylation between the NAFLD and healthy control groups, they found that hypomethylated CpG sites of acyl-CoA synthetase long-chain family member 4 (*ACSL4*) and carnitine palmitoyltransferase 1C (*CPT1C*) among 863 differentially methylated CpG sites were significantly associated with an increased risk of NAFLD [138]. They also reported 6 differentially methylated CpG sites between biopsy-proven simple hepatic steatosis and NASH from patients with NAFLD [139]. These genes included *ACSL4*, cardiolipin synthase 1 (*CRLS1*), *CPT1A*, single immunoglobulin and toll-interleukin 1 receptor domain (*SIGIRR*), single-stranded DNA binding protein 1 (*SSBP1*) and zinc finger protein 622 (*ZNF622*), and their DNA methylation levels were correlated with serum lipid contents, liver enzyme or liver histological parameters. Notably, in both analyses, hypermethylation at *ACSL4* (cg15536552) was shown to increase susceptibility to NASH, suggesting it as a potential biomarker for NAFLD progression. A genome-wide association study of peripheral blood DNA methylation correlated with hepatic fat accumulation was recently conducted in 3400 European ancestry, 401 Hispanic ancestry and 724 African ancestry participants from 4 population-based cohorts [140]. It was found that 22 CpGs were associated with hepatic fat, particularly in European ancestry participants, and that hypomethylation of a specific CpG site at the long intergenic nonprotein coding RNA 649 (*LINC00649*) gene was significantly associated with NAFLD and risk for new-onset type 2 diabetes (T2D). This result implies the presence of racial disparities in the regulation of DNA methylation during hepatic pathogenesis.

However, alterations in DNA methylation due to the prevalence of disease need to be discriminated from the DNA methylation changes associated with aging. There are algorithms that predict human chronological and biological age based on DNA methylation [141,142,143,144,145,146]. Different levels of DNA methylation from the level of predicted DNA methylation for a defined chronologic age represent age acceleration [142]. Loomba et al. [147] examined the Horvath clock, one of these algorithms derived from the methylation levels of 353 discrete CpG sites, including 193 hypermethylated and 160 hypomethylated CpGs, which were associated with age [142], in peripheral blood from patients with NASH. The results showed that age acceleration assessed by DNA methylation was observed among patients with NASH with fibrosis stages of F2–F3 compared with the control groups. Age acceleration in patients with NASH correlated with the hepatic collagen content assessed by quantitative morphometry of collagen staining, although it did not differ by fibrosis stage. In addition to NAFLD, the epigenetic clock can be accelerated in diseases and metabolic insults, such as human immunodeficiency virus and obesity, which are associated with increased liver fibrosis compared with age-matched controls [148,149,150].

It, however, still challenges to use cfDNA as a biomarker because of unclear origin and low concentration of cfDNA in plasma. MethyLigh droplet digital PCR, a recently developed technology, overcomes these limitations by amplifying the amount of cfDNA and inferring the cells/tissues origin of cfDNA based on the quantification of tissue-specific methylation patterns [133,151]. The U.S. Food and Drug Administration-approved blood test for detecting DNA methylation utilizing the method is currently available for colorectal cancer [152]. Thus, more and further studies are required to establish a big database of liver- and liver-cell-type-specific DNA methylation patterns to improve the accuracy and sensitivity of cfDNA as a potential biomarker for NAFLD.

### 4.3. Dysregulation of the DNA Methylation Process

DNA methylation and demethylation are a series of chemical reactions catalyzed by multiple enzymes, such as DNMTs and TETs, respectively [52,63]. Alterations in the expression and/or activity of these enzymes can affect global or locus-specific DNA methylation. In a mouse model, changes in hepatic DNMT1 and DNMT3A levels were shown to be associated with the development of hepatic steatosis [153]. Additionally, in humans, it was reported that the expression of three DNA methyltransferases, namely, DNMT1, DNMT3A and DNMT3B, increased in fibrotic livers [25]. In contrast, the hepatic expression of DNA demethylases, i.e., TETs, was downregulated in chronic liver disease. Both changes made differences in the global 5-mC and 5-hmC levels that resulted in genome-wide alterations in transcription [25]. A recent study investigated the role of 25-hydroxycholesterol (25HC), an oxidized product of cholesterol called oxysterol, in the epigenetic regulation of lipid accumulation in hepatocytes induced by high glucose (HG) in cell culture medium [154]. Oxysterols act as ligands for receptors that function in lipid metabolism and inflammatory responses, and they are known to play a role in the pathogenesis of several diseases, including NAFLD and other MetS [155,156]. The results showed that accumulated 25HC in the nuclei of hepatocytes cultured under HG conditions specifically activated DNMT1 and significantly upregulated 5-mC levels in at least 2225 genes involved in 57 signaling pathways, such as the phosphoinositide 3 kinase (PI3K), cyclic adenosine monophosphate (cAMP), insulin, diabetes and NAFLD pathways [154]. It is known that 25HC activates the liver X receptor (LXR)/SREBP-1C signaling pathway, which upregulates the biosynthesis of free fatty acids and triglycerides [157]. The acetyl-CoA surplus produced from a high-sugar diet can be used for the synthesis of cholesterol and subsequent oxysterols such as 25HC [158], which might explain how a high-sugar diet exerts adverse effects on the liver.

DNMTs were also present within mitochondria. Bellizzi et al. [159] analyzed the methylation status of mtDNA on human and mouse DNA isolated from blood and cultured cells by bisulfite sequencing and immunoprecipitation assays for 5-mC and 5-hmC. 5-mC in mtDNA was particularly observed within non-CpG nucleotides (cytosines followed by adenine, thymine, or another cytosine) mostly in the promoter region of the heavy strand and in conserved sequence blocks, suggesting that mtDNA methylation play a role in the regulation of mtDNA replication and/or transcription. A conflicting result was that inactivation of DNMTs in mouse embryonic stem cells reduced CpG methylation, while non-CpG methylation was unaffected, indicating that there might be a distinct pathway that regulates methylation for non-CpG sites. There is evidence that mtDNA methylation can be altered during the development of NAFLD [116,117,128,160].

Genetic variations in TET enzymes may also contribute to NAFLD progression. Pirola et al. [127] demonstrated that *TET1* rs3998860, the missense p.lle1123Met variant of *TET1*, was significantly associated with the serum levels of the CK18 fragment, a cell death biomarker, and disease severity in patients with NAFLD. *TET2* rs2454206, the substitution p.lle1762Val variant of *TET2*, was found to be associated with hepatic *PPARGC1A* methylation and its gene expression and the presence of T2D. Interestingly, Pirola et al. unexpectedly found that the global amounts and quantified nuclear staining levels of hepatic 5-hmC between patients with NAFLD and normal controls did not differ. Nevertheless, patients with NAFLD had significantly lower levels of cytosolic 5-hmC staining than normal controls, suggesting that aberrant 5-hmC expression might contribute to the pathogenesis of NAFLD by regulating mitochondrial biogenesis and PPARGC1A expression in the liver.

The methyl-CpG-binding domain proteins also play an important role in the regulation of epigenetic phenotypes and their relevant gene expression [49,52]. The methyl-CpG-binding domain proteins, such as MeCP2 and MBDs, read DNA methylation in cells and then recruit transcriptional repressor complexes to the methylated sites, which therefore means that they are considered transcriptional repressors. It has been reported that HSCs begin to express MeCP2 and MBD1-4 during activation to become myofibroblastic HSCs [28,29,161,162]. MeCP2 has been identified as a transcriptional repressor of the *PPARγ* gene, a master transcriptional regulator of the quiescent phenotype of HSCs [28,29,163]. It was also reported that the profibrogenic phenotype of activated HSCs is suppressed by the inhibitor of DNMT, 5′-aza-deoxycytidine [29]. MeCP2 has been reported to have the ability to not only transcriptionally inhibit but also to activate gene expression, although the underlying mechanisms are not yet fully understood [164]. Mice lacking MeCP2 are resistant to the development of fibrosis after chronic liver injury by which MeCP2 normally stimulates transcription of multiple fibrogenic genes [28]. In both ways, MeCP2 functions as a master epigenetic regulator of the myofibroblastic phenotype of activated HSCs and hepatic fibrogenesis.

## 5. Conclusions

Recent evidence from human and animal studies clearly shows that the pathogenesis of NAFLD is under both genetic and environmental regulation. In this review, we specifically discussed how the DNA methylation processes are involved in the prevalence and progression of NAFLD. It has been reported that tissue-specific DNA methylation occurs not only in nuclear genomic DNA but also in mtDNA. Diverse human studies have demonstrated that altered DNA methylation at global and particular CpG sites is observed with inverse correlations with pertinent gene expression. It is noted that DNA methylation or demethylation status is reversible, especially by methyl donor supplementation. It seems that simply changing from HF, HFS, WD or HFr-diet to a normal chow diet in mice and losing weight following bariatric surgery can also partially normalize DNA methylation profiles in NAFLD. Hence, the DNA methylation signature holds potential as a therapeutic target as well as a biomarker capable of staging NAFLD that can be utilized for diagnosis, prognosis and treatment in clinical practice.

Nonetheless, much of the work remains to be done in establishing cost-effective therapeutic and diagnosis strategies for NAFLD by utilizing DNA methylation. It is challenging, as most of the studies have been conducted on small sample sizes to discover DNA methylation sites associated with severe histological scores of NAFLD with high specificity and accuracy. To overcome this, researchers have tried meta-analysis studies involving multiple population-based cohorts. Although The Cancer Genome Atlas (TCGA) could be employed in a study of HCC-associated DNA methylation, only 3% of deposited HCC samples in the TCGA database have NAFLD as the background etiology [165]. In addition, there is a methodological shortcoming; for example, the widely used bisulfite sequencing is unable to distinguish between 5-mC and 5-hmC [166,167]. The development of a CRISPR/Cas9-targeting method for the experimental manipulation of sequence-specific DNA methylation could improve our understanding of the mechanism of DNA methylation-mediated gene expression regulation [168,169]. Finally, future studies are warranted to explore the mechanisms behind the close relationship between DNA methylation and gene expression in the liver. It would be important to comprehend the complex interactions of DNA methylation with other epigenetic modifications, such as histone modifications and noncoding RNAs, and the interactions between various liver cell types that may each have distinct DNA methylomes [134].

In conclusion, a better understanding of the DNA methylation mechanisms operating during the pathogenesis of NAFLD will potentially give rise to diagnostic, prognostic, and therapeutic interventions, promising great advances in personalized medicine.

## Figures and Tables

**Figure 1 ijms-21-08138-f001:**
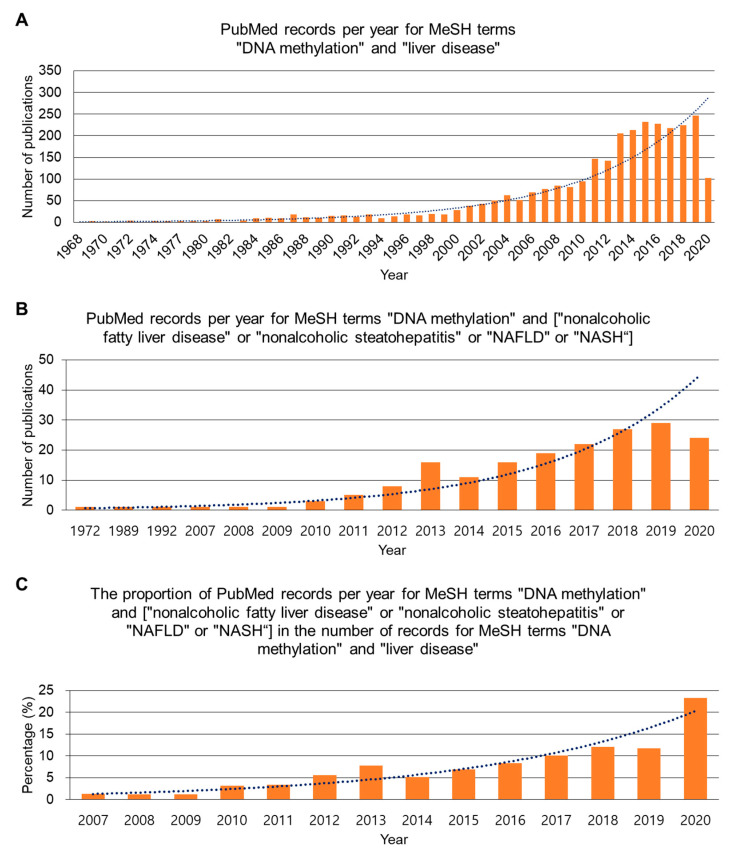
The number of publications on DNA methylation in nonalcoholic fatty liver disease/nonalcoholic steatohepatitis (NAFLD/NASH) is increasing. (**A**) The number of publications in PubMed per year for the terms of medical subject headings (MeSH), “DNA methylation” and “liver disease”. *X*-axis scale is displayed every 2 years. (**B**) The number of publications in PubMed per year for the terms of MeSH, “DNA methylation” and (“nonalcoholic fatty liver disease” or “nonalcoholic steatohepatitis” or “NAFLD” or “NASH”). (**C**) The proportion of PubMed records per year for MeSH terms shown in panel B in the number of PubMed records for MeSH terms shown in panel A. The records until September 2020 were included.

**Figure 2 ijms-21-08138-f002:**
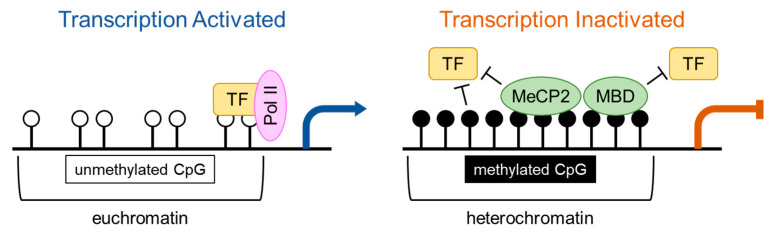
DNA methylation mediates gene silencing. In euchromatin (loose or open chromatin), cytosine-phospho-guanine dinucleotide (CpG) sites are generally unmethylated in promoter region and accessible to transcription factors (TFs). Methylation of cytosine in CpG sites is associated with heterochromatin (tight or closed chromatin) that typically results in silencing of gene transcription. Methylated CpG prevents the binding of TFs directly or indirectly by interacting with readers of DNA methylation, such as methyl CpG binding protein-2 (MeCP2) and methyl-CpG-binding domain (MBD). Pol II indicates DNA-dependent RNA polymerase II.

**Figure 3 ijms-21-08138-f003:**
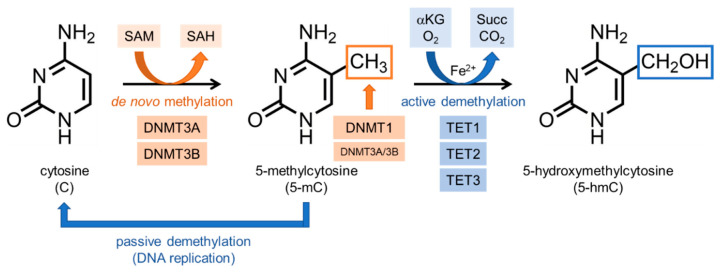
Enzyme-mediated DNA methylation and demethylation processes. DNA methylation occurs at the carbon 5 of cytosine in CpG dinucleotides. DNA methyltransferases (DNMTs) catalyze the methylation reaction by transferring a methyl group to cytosine (C) using S-adenosyl methionine (SAM) as methyl donor and producing S-adenosyl homocysteine (SAH). As a product, 5-methylcytosine (5-mC) is generated. DNMTs also maintains the status of DNA methylation. DNA demethylation is a multi-step oxidation process mediated by ten-eleven translocation (TET) methylcytosine dioxygenases (active demethylation). TET enzymes use Fe^2+^ and α-ketoglutarate (αKG) as co-substrates, and generates succinate (Succ) and CO_2_. In the first step of demethylation process, the 5-mC is converted to 5-hydroxymethylcytosine (5-hmC) (shown), and further TET-dependent oxidation leads to the conversion of 5-hmC into 5-formylcytosine (5-fC) and 5-carboxylcytosine (5-caC) (not shown). After several oxidation reactions mediated by TET enzymes, the methyl group is removed by base excision repair mechanism. 5-mC or 5-hmC can also be converted to unmethylated cytosine during DNA replication, through a passive DNA demethylation.

**Table 1 ijms-21-08138-t001:** Alteration of hepatic DNA methylation in NAFLD patients.

Gene Name (CpG Site)	Alteration	Sample Type	Correlations with DNA Methylation	Number of Subjects	DNA Methylation Measurement Method(s)	Year [Reference]
*TRIM4* (cg02756107, cg25591451), *PRC1* (cg01407062), *TUBA1B* (cg13709639)	Hypo	PHx specimen	NASH-HCC specific, necroinflammatory grade of NASH, poorer tumor differentiation	36 controls, 22 NASH-HCC (paired cancerous and non-cancerous tissues), 36 viral-HCC (HBV and/or HCV)	Infinium HumanMethylation450 BeadChip	2020 [126]
*WHSC1* (cg03150409), *MAML3* (cg14497545), *WDR6* (cg13003239), *FLCN* (cg07138452)	*WHSC1*, *MAML3*: hypo; *WDR6*, *FLCN*: hyper	Liver biopsy or PHx specimen	NASH-HCC specific	55 controls,113 NASH, 22 NASH-HCC, 37 viral hepatitis or cirrhosis (13 HBV, 23 HCV, 1 both), 37 viral-HCC	Infinium HumanMethylation450 BeadChip	2017 [125]
*HIC1*, *GSTP1*, *SOCS1*, *RASSF1*, *CDKN2A*, *APC*	Hyper	Paraffin-embedded liver biopsy	8-OHdG	65 NAFLD	MethyLight assay	2016 [118]
*FADS2* (cg06781209, cg07999042)	Hypo	Liver biopsy	D6D activity, *FADS2* rs174616 ^1^ (cg07999042)	95 obesity	Infinium HumanMethylation450 BeadChip	2019 [120]
*BCAT1* (cg09800500, cg07479001, cg16490209)	Hypo	Frozen liver biopsy	Poor outcome ^2^	86 NAFLD ^3^ (47 mild and 39 advanced fibrosis)	Infinium HumanMethylation450 BeadChip	2018 [123]
*APO* genes, *NPC1L1*, *STARD*, *GRHL*	*APO*, *NPC1L1*, *STARD*: hyper; *GRHL*: hypo	Liver biopsy	NAFLD, liver fibrosis	75 controls, 103 NAFLD	Infinium HumanMethylation450 BeadChip	2017 [122]
*CYP27A1*, *SLC51A*, *SLC27A5*, *SLCO2B1*, *SLC47A1*, *UGT* and *CYP genes*	*CYP27A1*, *SLC51A*, *SLC27A5*, *SLCO2B1*, *SLC47A1*: hyper; *SLCO2B1*: hypo	Liver biopsy	NAFLD, liver fibrosis	75 controls, 103 NAFLD	Infinium HumanMethylation450 BeadChip	2016 [121]
*FGFR2* (cg01385327), *CASP1* (cg13802966), *MAT1A* (cg19423196)	*FGFR2*, *CASP1*: hypo; *MAT1A*: hyper	Frozen liver biopsy	Liver fibrosis	25 controls, 90 NAFLD (52 mild and 38 severe fibrosis)	Infinium HumanMethylation450 BeadChip, bisulfite pyrosequencing	2013 [114]
30 CpG sites ^4^ including *LDHB* (cg04949489) and *SQSTM1* (cg08836954)	N/A	Liver biopsy	NASH, serum fasting insulin level	95 obesity (34 normal, 35 SS and 26 NASH)	Infinium HumanMethylation450 BeadChip	2017 [113]
*PC* (cg04174538), *ACLY* (cg25687894), *PLCG1* (cg18347630), *IGF1* (cg08806558), *IGFBP2* (cg11669516), *PRKCE* (cg04035064), *GALNTL4* (cg16337763), *GRID1* (cg27317356), *IP6K3* (cg10714061)	*PC*, *IGF1*, *IGFBP2*, *GRID1*: hyper; *ACLY*, *PLCG1*, *PRKCE*, *GALNTL4*, *IP6K3*: hypo	Frozen liver biopsy	NAFLD	18 controls, 45 obesity (18 healthy obese, 12 SS and 15 NASH; including 23 post-bariatric patients)	Infinium HumanMethylation450 BeadChip, bisulfite sequencing	2013 [31]
Global DNA methylation	Hypo	Liver biopsy	Inflammation, liver fibrosis, disease progression, serum homocysteine level	18 controls, 47 NAFLD	MethylFlash Methylated DNA Quantification Kit ^5^	2020 [124]
*PPARα*, *PPARγ*, *TGFβ1*, *PDGFα*	*PPARα*, *PPARγ*: hyper; *TGFβ1*, *PDGFα*: hypo	Paraffin-embedded liver biopsy	Liver fibrosis	17 NAFLD (8 mild and 9 severe fibrosis)	Bisulfite pyrosequencing	2015 [30]
*MT-ND6*	Hyper	Liver biopsy	NAS, liver fibrosis	18 controls, 45 NAFLD (23 SS and 22 NASH)	MS-PCR	2013 [116]
*PPARGC1A*	Hyper	Liver biopsy	Plasma fasting insulin level, HOMA-IR	11 controls, 63 NAFLD	MS-PCR	2010 [115]
*DPP4*	Hypo	Frozen liver biopsy	Hepatic steatosis	96 obesity	Bisulfite pyrosequencing	2017 [98]

^1^*FADS2* SNP G > A; ^2^ Cardiovascular outcomes and/or hepatic decompensation; ^3^ Gene expression and clinical outcome data were available in 55 cases; ^4^ Genes of which DNA methylation and mRNA expression were significantly correlated; ^5^ Fluorescence-based quantification of global DNA methylation (5-mC) in an ELISA-like format (EpiGentek Group Inc., Farmingdale, NY, USA). Hypo, hypomethylation; Hyper, hypermethylation; PHx, partial hepatectomy; 8-OHdG, 8-hydroxydeoxyguanosine; D6D, delta-6 desaturase; BA, bile acid; DM, drug metabolism; N/A, not available; SS, simple steatosis; NAS, NASH activity score; MS-PCR, methylation-specific PCR; HOMA-IR, homeostasis model assessment of insulin resistance.

**Table 2 ijms-21-08138-t002:** Alteration of circulating DNA methylation in NAFLD patients.

Gene Name (CpG Site)	Alteration	Sample Type	Correlations with DNA Methylation	Number of Subjects	DNA Methylation Measurement Method(s)	Year [Reference]
152 CGIs	120 CGIs: hyper; 32 CGIs: hypo	Peripheral blood	Age acceleration, liver fibrosis	18 controls, 44 NASH (15 F2 and 29 F3)	Infinium HumanMethylation450 BeadChip	2018 [147]
22 CpG sites including *LINC00649* (cg08309687)	17 genes including *LINC00649*: hypo; 5 genes: hyper	Peripheral blood	Hepatic fat, risk for NAFLD and new-onset T2D (cg08309687)	3400 EA, 401 HA, 724 AA	Infinium HumanMethylation450 BeadChip	2019 [140]
*ACSL4* (cg15536552), *CRLS1* (cg05131957), *CPT1A* (cg00574958), *SIGIRR* (cg13463639), *SSBP1* (cg12473838), *ZNF622* (cg16398128)	*ACSL4*: Hypo ^1^	Peripheral blood	NASH	30 controls, 35 NAFLD (18 SS and 17 NASH)	Infinium HumanMethylation450 BeadChip, bisulfite pyrosequencing	2018 [139]
*ACSL4* (cg15536552), *CPT1C* (cg21604803)	Hypo ^2^		Risk for NAFLD, susceptibility to NASH (cg15536552)			2018 [138]
*SLC7A11*(cg06690548)	Hypo	Peripheral blood	Hepatic steatosis, serum liver enzymes (GGT, ALT)	1450 participants (731 from discovery cohort, 719 from replication cohort)	Infinium HumanMethylation450 BeadChip	2017 [137]
*PPARγ*	Hyper ^3^	Plasma	Liver fibrosis	9 controls, 26 NAFLD (14 mild and 12 severe fibrosis), 13 ALD cirrhosis	Bisulfite pyrosequencing	2017 [136]

^1^ This was validated by bisulfite pyrosequencing; ^2^ After the adjustment of age, gender, body mass index and homeostasis model assessment of insulin resistant; ^3^ This was not NAFLD-specific and similar results were found in ALD cirrhosis. CGIs, CpG island; Hyper, hypermethylation; Hypo, hypomethylation; F, fibrosis stage; EA, European ancestry; HA, Hispanic ancestry; AA, African ancestry; SS, simple steatosis; GGT, gamma-glutamyl transferase; ALT, alanine aminotransferase; ALD, alcoholic liver disease.

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
