# Peer review of "DNA Methylation in Nonalcoholic Fatty Liver Disease"

_ijms, 2020, doi:10.3390/ijms21218138_

Round 1
Reviewer 1 Report
The paper from Hyan & Jung describes some relevant apects in detecting and monitring the development of NAFLD and NASH based on the detection of methylation processes that can affect gene expression which are influenced by environmental and lifestyle experiences such as diet, obesity, and physical activity and are reversible.
The approach is novel and the most relevant advances in this field are well described. References are updated.
On the basis of these conclusion, I believe that the paper is suitable of publication in the present form.
Author Response
Thank you so much for your time and generosity.
The manuscript has been proofread by professional profreading company. If you request, I will submit the editing certificate.

Reviewer 2 Report
Hyun and Jung have submitted an excellent review and perspecctive article on DNA methylation in NAFLD. With only one exception, I find the manuscript outstanding. That exception is the discussions of DNA methylation as a biomarker. These discussions occur in the abstract and at several other places in the manuscript. They are not well grounded and I find them misleading. Certainly, technologies will continue to advance and perhaps this will become feasible some day, but to my knowledge, the only way this could have diagnostic utility with current technologies would require sample collection by a liver biopsy procedure. Once a biopsy is taken, it is unclear how sequencing (or other technologies) to determine DNA methylation status would be advantageous over histology, IHC, transriptome analysis, etc. The authors allude to being able to use circulating DNA, and cite studies on methylation studies of circulating leukocytes; however they do not adequately address the technological hurdles that prevent this from being feasible and accurate for assessing hepatocyte health-status. With leukocytes, it is straight-forward to address cell identity and quality. With circulating DNA, current technologies cannot likely reliably identify the cell-type it came from. Moreover, if one could use epigenetic fingerprints to distinguish hepatocyte DNA from other possible DNA sources in circulation, it would not be coming from representative hepatocytes that are surviving and contributing to liver physiology. Rather, it would be from a non-representative subset of cells that died - probably from necrotic death in order to have released DNA. Similarly, if whole hepatocytes were released into the circulation (unlikely, since this would likely be damaging to small capillaries throughout the body), these would not be representative of the status of the surviving cells in the liver.
Summary: This is an outstanding review of DNA methylation in liver and should be published with high priority after the unfounded discussions of using hepatocyte DNA methylation as a biomarker are removed or presented in a more realistic context regarding the current technologies available.
Author Response
Thanks you for your comments. Your critique is reasonable, and we totally agree with your comment. As you pointed out, we revised the manuscript; we deleted several parts that emphasized the use of hepatocyte DNA methylation as a biomarker. Rather, we have discussed that hepatocyte DNA methylation has a “potential” to be developed as a biomarker for NAFLDs, but more efforts are still needed to overcome current poor knowledge and technological hurdles. Changed parts are marked with track change for your convenience.
- Lines 17-19: a sentence was inserted (“Poor understanding…therapies and biomarkers.”).
- Lines 19-22: a sentence was removed (“Given its increasing…treatment for NAFLD.”).
- Line 26: the words (“that can determine”) were replaced with another word (“indicating”).
- Line 27: the words (“therapeutics for NAFLD”) were added, and a word (“prognosis”) was deleted. Also, the words (“In this review”) were replaced with a word (“Herein”).
- Line 28: a word (“provide”) was replaced with another word (“review”).
- Lines 29-30: a word (“novel”) was replaced with another word (“potential”) and two words (“treatments and”) were added.
- Lines 55-56: a sentence was edited (“biomarkers to diagnose and stage” à “strategies for treating and staging”).
- Line 73: the words (“therapeutic targets and”) were inserted.
- Line 284: a typo (“Dna”) was corrected (“DNA”).
- Line 399: the sentence was edited (“diagnosis and prognosis” à “diagnosis, prognosis and therapeutic intervention”).
- Line 406: a typo (“Dna”) was corrected (“DNA”).
- Lines 407-412: three sentences were deleted (“As discussed above, DNA methylation…procedures that carry the aforementioned limitations.”).
- Lines 417-418: a sentence was deleted (“Thus, the level of DNA methylation…such as liver biopsy.”).
- Lines 418-419: a paragraph (“Hardy et al. [136] …DNA methylation during hepatic pathogenesis.”) was placed right after the front sentence (“Given that hepatocyte death…with high accuracy to stratify NAFLD [136].”).
- Line 452: the first word in the sentence was edited (“Alterations” à “However, alterations”).
- Lines 466-473: four sentences were added (“It, however, still challenges to use cfDNA…as potential biomarker for NAFLD.”).
- Line 537-538: the sentence was edited (“signature could be utilized…of disease progression” à “processes are involved…progression of NAFLD”).
- Line 545: several words were inserted (“therapeutic target as well as”).
- Line 546: the sentence was edited (“diagnosis and prognosis” à “diagnosis, prognosis and treatment”).
- Line 548: two words were inserted (“therapeutic and”).
- Line 549: a word was edited (“strategy” à “strategies”).